# Addressing bias in Face Detectors using Decentralised Data collection with incentives

**Ahan M R**
BITS Pilani, Goa Campus
Bangalore
India
f20160487@goa.bits-pilani.ac.in

**Robin Lehmann**
DataUnion Foundation
160 Robinson Road
#14-04 Singapore Business Federation Centre
Singapore
robin@dataunion.app

**Richard Blythman**
Algovera
Dublin
Ireland
richard@algovera.ai

## Abstract

Recent developments in machine learning have shown that successful models do not rely only on huge amounts of data but the right kind of data. We show in this paper how this data-centric approach can be facilitated in a decentralised manner to enable efficient data collection for algorithms. Face detectors are a class of models that suffer heavily from bias issues as they have to work on a large variety of different data.

We also propose a face detection and anonymization approach using a hybrid Multi-Task Cascaded CNN with FaceNet Embeddings to benchmark multiple datasets to describe and evaluate the bias in the models towards different ethnicities, gender and age groups along with ways to enrich fairness in a decentralized system of data labelling, correction and verification by users to create a robust pipeline for model retraining.

## 1 Introduction

The amount of data available and used for training public datasets is vast, yet there is an inherent bias in these datasets towards certain ethnicity groups like caucasian faces as compared to other ethnicities such as Asian, African, Indian, etc. There is definitely a need to mitigate the bias and emphasize on the improvement of fairness in face detection algorithms. This will improve the efficiency and accuracy of Face Verification (FV), recognition, anonymization and other use-cases of face detection.

With the advent of publicly available images on social media and the internet, there is a need to enforce personal privacy of people by performing face anonymization on these images. In this work, we propose a ML pipeline to detect faces using a robust multi-task cascaded CNN architecture along with other pre-trained models such as VGGFace2 [3] and FaceNet [15] to anonymize the detected faces and blur them using a Gaussian function. We also benchmark the performance of certain custom and pre-trained models on various open-sourced datasets such as MIAP [16], FairFace [9], and RFW [21] (Racial Faces in Wild) to understand the bias of models trained on these datasets. Along with face anonymization, we also determine the age and gender demographics of the detected faces to find any bias present in open-source models. We also evaluate the performance of these open-source models before and after training it on a diverse and fairness-induced dataset by proposing

36th Conference on Neural Information Processing Systems (NeurIPS 2022).

a decentralized system of data evaluation and verification by users of the model output generated (faces detected in the input), see section 3.3.

Lastly, we also discuss ways to de-bias the data during pre-processing and post-processing and how to reduce the false positives using clustering and statistical analysis of the generated output. We propose a decentralized platform for data collection and annotation of data with user incentives for detecting any machine-undetected faces in images as part of an initiative to increase model fairness and reduce ethnicity, age, and gender bias.

## 2 Related Work

The current systems in computer vision have higher yield and astonishing results in several areas, but there are several societal issues related to demographics, ethnicity, gender, age, etc. that have been discussed more recently due to their usage in face recognition, object detection and other applications [18] [19] [8]. Most image recognition algorithms have high disparity in performance on images across the world as discussed in [5] [17] [22] due to the bias in the dataset used for training and also the differences in pipeline used. This bias is generally due to dataset disparity since most of the open-source datasets created and benchmarked are localized to only a few locations restricting the diversity in data quality. Secondly, the other set of related papers talk about harmful and mislabelled data associations which can often lead to a lot of wrongful associations across gender and ethnicity groups in general as discussed by Crawford et al. [4]. Some of the other indicators which causes disparity in performance of a face detection algorithm towards certain groups of people is due to bias in learned representations or embeddings of users of underrepresented groups and other demographic traits. Raji et al. [14] talks about reduction of errors in evaluating the commercial face detectors by changing the evaluation metrics used. Ensuring privacy as part of face recognition campaign is an equally important problem, and limited research has been done on the task of extracting and remove private and sensitive information from public dataset and image databases. There has been a few previous work done in literature [2] [12] [7] that blur the background or use gaussian/pixelation functions to blur faces in an image.

To improve the robustness and add fairness to the datasets and models used in the above problem approach, we propose a decentralized tool for collecting, annotating and verifying the face detections made by face recognition algorithms across different parts of the world to ensure the data samples collected are rich in diversity, help identify the bias in current commercial and open-source models, generate edge-cases and training samples that could be used to retrain these detectors to improve the coverage of data distribution learnt by our models.

## 3 Methodology

We aim to build a robust face anonymization pipeline along with functionalities to determine the characteristics of the detected faces as shown in Fig 1 on a decentralized platform for verification and annotation. We also try to estimate the bias towards certain ethnicities and characteristic features in some of the popular pre-trained model architectures such as MTCNN (Multitask Cascade CNN) [23], FaceNet [15] and RetinaNet [10] against the open-source datasets used for understanding and evaluating bias in the face detectors.

### 3.1 Datasets

In order to understand the bias of ethnicity, age, and gender, it is important to evaluate the performance of classification of different ethnicities as a binary task of faces detected and undetected to understand if there is a bias towards some ethnicity classes having stronger attribute indicators as compared to the rest. The following datasets are a good benchmark to determine the bias since each of these datasets have been labelled and open-sourced keeping the diversity and inclusion of most ethnicities in mind.

**MIAP Dataset:** The MIAP (More Inclusive Annotations for People) Dataset [16] is a subset of Open Images Dataset with new set of annotations for all people found in these images enabling fairness in face detection algorithms. The Dataset contains new annotations for 100,000 Images *(Training set of 70k and Valid/ Test set of 30k images)*. Annotations of the dataset includes 454k bounding boxes along with Age and Gender group representations.

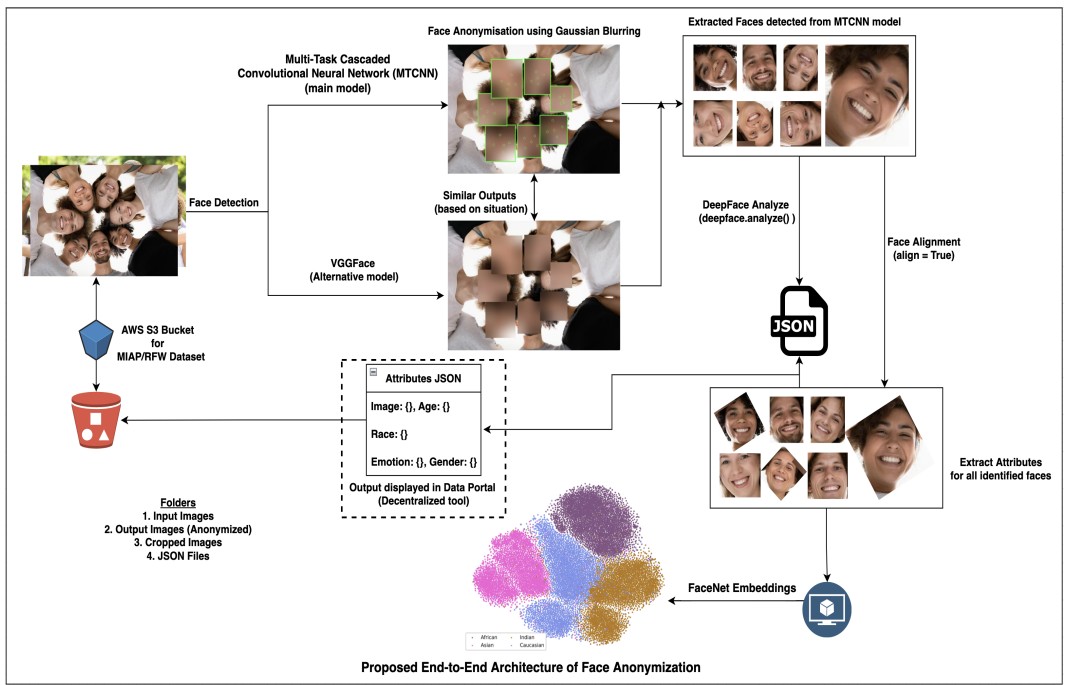

Figure 1: End to End Architecture of Face Anonymization and attribute extraction

**FairFace Dataset:** FairFace[9] a facial image database contains nearly 100k images which is also created to reduce the bias during training by having equal representation of classes from YFCC-100M Flickr dataset [20]. The dataset consists of 7 classes namely, White, Latino, Indian, East Asian, Southeast Asian, Black and Middle Eastern. Models trained on FairFace have reported higher performance metrics[9] as compared to other general datasets, and hence, we have included this dataset as well as part of our study.

**Racial Faces in the Wild (RFW):** The RFW [21] Database primarily consists of four test subsets in terms of ethnicity backgrounds, namely Indian, Asian, African and Caucasian. Each subset consists of images for face verification, which is around 10k images of 3k individuals.

## 3.2   Architecture

The end-to-end pipeline uses multiple models to detect faces from the input images. MTCNN [23] and VGGFace [3] are used for generating bounding boxes of the detected faces, post which, we enhance the output bounding boxes and extract the face image to generate a gaussian blurred image as part of our goal to anonymize the faces. These architectures have been employed and are chosen as standard models for face attribute extraction algorithms. The non-anonymized copy of the detected images are used as input to the FaceNet [15] model for generating the face embedding vectors.

The **MTCNN** architecture proposed by Zhang et al. [23] mainly consists of three different stages and each stage consists of a Neural Network, namely, the Proposal Network, Refine Network and the Output network. The first stage uses a shallow CNN architecture to generate candidates proposal windows, which the Refine network enhances with a deeper CNN. The output network refines the result of previous layers and generates the face landmark positions. Since the architecture uses different face landmark locations to estimate a face, we use it as part of our experiment to evaluate face recognition datasets for estimating inherent bias.

**FaceNet** is another model proposed by Schroff et al. [15] outputting a 128-dimension vector, also known as a face embedding which is optimized to differentiate between similar and dissimilar faces using euclidean metrics. The architecture uses a triplet-based loss function, which uses positive and negative samples to estimate the distance between each other respectively as part of the loss function. For each face detected in the inferred image, an embedding is calculated. We use FaceNet embeddings

to cluster similar faces using DBSCAN [6] on faces extracted from the MTCNN model. DBSCAN generally uses two parameters, namely, the minimum distance between two instances for them to be grouped together, and second, the number of points to form a cluster. So, if distance between two faces is high, they tend to form different clusters. PCA [11], a very popular dimensionality reduction technique is used to reduce the 128-dimensional vector to 2-dimensional vector to visualize the cluster faces as part of estimating bias in the algorithms.

Generally the undetected faces and misclassified faces in the dataset for different pre-trained and popular model architectures form outliers or belong to wrong clusters which are easy to identify. We then employ clustering metrics to estimate the embedding based partitioning of clusters, such as: *Mean Silhouette Coefficient* which measures similarity and dissimilarity between elements of a certain cluster, and, *Davies-Bouldin index* [13].

### 3.3 Decentralized Data Collection Platform

We propose a decentralized data platform for crowdsourcing images and image datasets. The purpose of the tool is to run inferences on images that users upload and perform face anonymization using our algorithm. It is important to note that, to curate a more diverse dataset as part of our initiative, the focus is added to incentivize users who upload and annotate images of the requested categories in terms of ethnicity and gender only. This creates two opportunities for incentivizing users to use our tool:

(**1**) Users can now annotate the requested category of diverse images on the interface directly in case of any undetected faces (False negatives) and wrong detections (False positives) and post successful verification by verifiers, will be incentivized in form of bounties and revenue shares, (**2**) Users can upload, annotate, and verify annotations of images while still keeping the ownership of their data - they give a license to the platform to use it and in return get a share in the revenue their contributions create, i.e, any form of revenue generated using models built using the dataset will ensure that users receive a royalty for contributing to the dataset. Also, the missed edge cases (face detections and false positives) across various images will be collected in the system and will be used for retraining the face detectors in a periodic manner to improve the performance of the model and enrich fairness and reduce the inherent bias in the data and trained model.

Distributing ownership of the dataset across creators, annotators and verifiers will democratize the system of ownership without only one central party controlling it and the revenue/value generated by the built algorithms and datasets can flow back to the community directly. The trained model and inference algorithm will be published on decentralized algorithm marketplaces so it will be possible to run inference in decentralized computing environments to make downloading and copying the model impossible. For end-to-end workflow, refer to appendix 3.

## 4 Results

As seen in Table 1, we present a few statistical metrics to determine the fairness for different ethnicities in the RFW dataset using MTCNN [23] model and FaceNet embeddings. It is not directly evident from the results of one group getting better results consistently, but a clear pattern in the bias towards certain ethnicities became evident on deeper study. The prediction accuracy for Asian (A) and Black (B) groups were lower compared to Indian (I) and White (W). But, this is not enough to indicate the bias as there isn't a significant difference between different groups. However, Positive Predictive Value (PPV) and False Postive Rate (FPR) indicate higher confidence in White faces than other groups with a significantly lower False Positive Rate, and this pattern is also seen in PPV, as for white faces, the value is as high as 0.98 and compared to Asian group which is only around 0.78 indicating a higher precision rates in detecting white faces compared to other groups.

We also tried to quantify the similarity between users in a given cluster extracted and processed by FaceNet embeddings followed by dimensional reduction techniques for both MIAP and RFW datasets in Table 2. As we can see, the trend in mean silhouette score (MSC) is such that attribute with higher number of distinct clusters has a higher score, indicating higher similarity between elements to their own cluster. Clearly for MIAP [16], as we can see when we calculate the metrics for combined clusters of race and gender, the MSC score is higher than when calculated individually, indicating that the gender clusters in any race are closer and correlated than between two ethnicities (racial

Table 1: Statistical metrics for RFW Dataset using pre-trained MTCNN + FaceNet embeddings

| Metrics (M) | Asian(A) | Indian(I) | Black(B) | White(W) |
|---|---|---|---|---|
| Prediction Accuracy | 0.91 | 0.95 | 0.92 | **0.97** |
| False Positive Rate | 0.07 | 0.04 | 0.08 | **0.005** |
| False Negative rate | 0.05 | 0.08 | 0.04 | 0.14 |
| Positive Predictive Value | 0.78 | 0.93 | 0.82 | **0.98** |

groups). The Davies-Bouldin index also shows a very similar pattern for the MIAP dataset indicating that clusters are best seperated when combined than when clustered individually in the order: both clustered together, racial groups clustered together and finally clustered based on gender.

Table 2: Clustering metrics for RFW and MIAP Dataset using MTCNN + FaceNet embeddings

| Metrics | RFW-Race | MIAP-Race | MIAP-Gender | MIAP-Both |
|---|---|---|---|---|
| MSC | 0.12 | 0.16 | 0.09 | 0.19 |
| DBI | 4.21 | 3.89 | 6.47 | 3.64 |

These results clearly state the need for a trained model that is unbiased towards all ethnicities and gender groups. To enrich fairness in training, the MTCNN + FaceNet model was retrained on a FairFace [9], a balanced dataset with equal distribution of all ethnicities and gender groups with adjusted labels of Race similar to RFW and MIAP Datasets. The increase in prediction accuracy of classes ranged between 1% to 5.5%, and PPV showed an increase of upto 19% after retraining. This shows there was a clear improvement in performance of the model as shown in Table 3 indicating that, an unbiased dataset used for training along with a few data augmentation techniques can improve the model performance such that, the results are not biased towards any single gender or racial group.

Table 3: Statistical metrics for RFW Dataset using FairFace trained MTCNN + FaceNet embeddings

| Metrics (M') | Asian(A) | Indian(I) | Black(B) | White(W) |
|---|---|---|---|---|
| Prediction Accuracy | 0.96 | 0.95 | 0.97 | 0.98 |
| False Positive Rate | 0.01 | 0.01 | 0.008 | 0.005 |
| False Negative rate | 0.05 | 0.03 | 0.04 | 0.04 |
| Positive Predictive Value | 0.93 | 0.92 | 0.94 | 0.98 |

Hence, as proposed in Section 3.3, a Data Portal will be used for curation and publishing of various datasets with the support of annotators and verifiers. The incentive of ownership in dataset usage and also for labelling incorrectly detected faces and missed face detections on the tool also helps increase the engagement of users on the portal to challenge the model and receive bounty in return. This will help us to periodically retrain the face anonymization models on various edge-cases and improve fairness in these models in a decentralized manner.

## 5 Conclusion

In conclusion, we believe that measuring fairness in face anonymization algorithms is necessary to deploy technology that is unbiased and more inclusive to all the different ethnicities, gender and age groups. We proposed a decentralized tool to improve the quality of training datasets used in modelling face recognition algorithms by shifting focus onto identifying and quantifying "bias" in the core algorithm towards different groups and de-biasing it. The debiasing steps included both, creating a diverse dataset with better representation of most demographics and retraining all the layers of the core algorithm to allow the same model to be fine-tuned (Dense layers only) periodically based on the missed detections identified by the annotators and verifiers in the tool. The bias measurement framework was outlied in this paper.

As part of our analysis, we figured that most face detection algorithms are predominantly biased towards white faces across both MIAP and RFW datasets irrespective of the gender groups. Since, the

clustered embeddings of FaceNet model showed that clustering metrics were much higher when male and female faces were clustered together across all ethnicities than when clustered seperately. This indicates a need for diversity in the dataset across all ethnicities; it is more likely to be fairer when the dataset creation happens in a more decentralised manner and users across the world contribute in adding images, identifying missed detections of a certain demographic group or in validating the corrected output from a fellow user.

In future work, we will focus on answering the questions raised during the above discussion in terms of breaking down the clusters in more detail to help us interpret correlation between the data points which led the model to cluster certain points closer to each other. We also plan to make the Data Portal public with access to all users to ensure that the users will be able to upload their own data into the pipeline and also get incentivized based on the usage of their data from any algorithm that is built on top of their data. We also plan to improve the anonymization algorithm using a GAN based approach to ensure the data distribution of the anonymized face does not change completely. In addition, we also plan to integrate Spotify's Annoy [1] for indexing similar faces across the Data Portal to find similar images of users uploaded for denoising the uploaded data for duplicates.

# 6 Appendix

## 6.1 Clustering similar faces using FaceNet Embeddings

The visual representation of the RFW (Racial Faces in Wild) dataset faces clustered using dimensionality reduction technique: t-SNE in 2-dim space followed by DBSCAN algorithm (Converted from 128-dim vector generated by FaceNet representations or face embeddings). As seen visually, similar

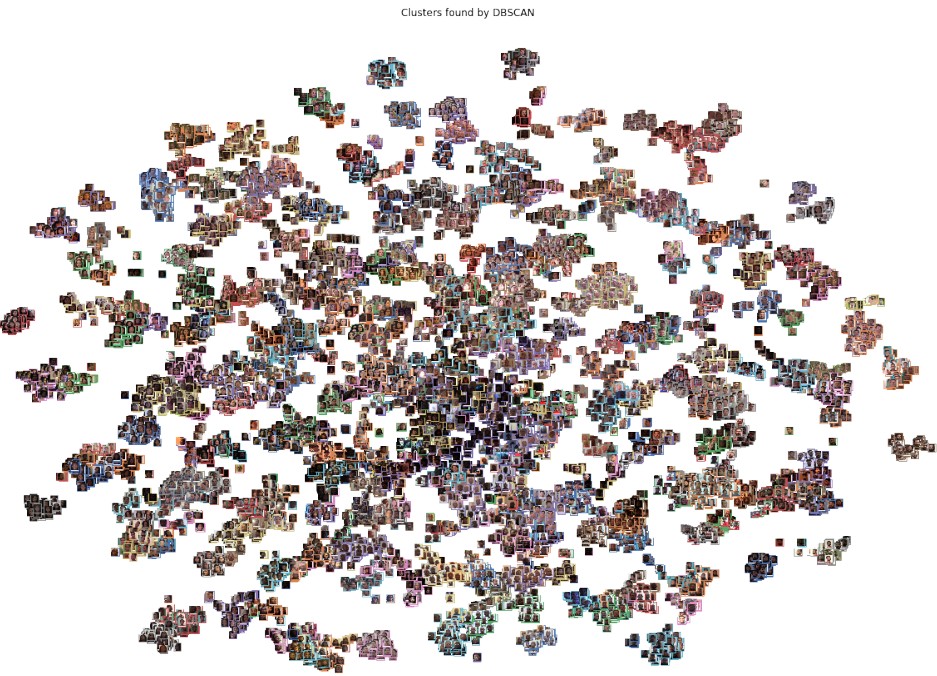

Figure 2: Face Embeddings visualized using t-SNE and DBSCAN

demographic groups are clustered closer to each other in the 2-D space. On using different clusters sizes, the density of clusters changed accordingly. The number of clusters that gave the optimal clustering metrics was chosen for benchmarking the dataset's clustering metrics.

## 6.2 Data Portal pipeline

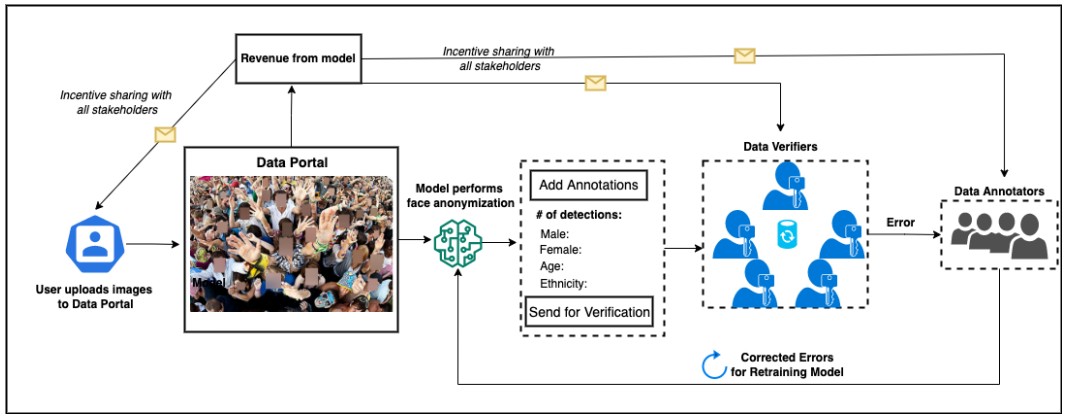

Figure 3: Proposed end-to-end working of the decentralized data portal

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
