# OpenReview forum: "Addressing Bias in Face Detectors using Decentralised Data collection with incentives"
_NeurIPS.cc/2022/Workshop/TSRML — TSRML2022_

### Official Review · Reviewer_fW2f · 2022-10-18
**Strong work on how bias affects certain ethnic groups and how to fix it**

**Overall Rating:** 10

**Summary:**

The authors propose a ML pipeline to detect faces using a robust multi-task cascaded CNN architecture to anonymize the detected faces and blur them using a Gaussian function. The work aims to evaluate the bias in the models towards different ethnicities, gender and age groups along with ways to ensure fairness. The authors also discuss ways to de-bias the data during pre-processing and post-processing and describe how to reduce the false positives using clustering and statistical analysis of the generated output.

**Strengths:**

The work was benchmarked against multiple well established pre-trained model architectures such as MTCNN (Multitask Cascade CNN), FaceNet and RetinaNet. The authors also ensured picking select datasets to test bias from the ones which have been labelled and open-sourced keeping the diversity and inclusion of most ethnicities in mind, such as The MIAP (More Inclusive Annotations for People), FairFace, and Racial Faces in the Wild (RFW) dataset.

The experiments demonstrate an important finding showing how the prediction accuracy for Asian and Black groups were lower compared to Indian and White. They also showcase how the Positive Predictive Value (PPV) and False Positive Rate (FPR) indicate higher confidence in White faces than other ethnic groups such as Asian, Black and Indian.

The authors also take steps to remove the above bias included creating a diverse dataset with better representation of most demographics and retraining all the layers of the core algorithm to allow the same model to be fine-tuned periodically based on the missed detections.

**Weaknesses:**

No inherent weakness in the work.

**Overall Recommendation:**

Overall, the paper was well articulated and showcased results with proper experimentations running across multiple well established architectures and datasets. The work is quite significant as it not only highlights the bias towards a certain ethnic group (White) and also goes above and beyond to build a framework to reduce the bias.

**Review Confidence:**

4: The reviewer is confident but not absolutely certain that the evaluation is correct

---

### Official Review · Reviewer_ts9S · 2022-10-19
**Questionable idea effectiveness and writing issues**

**Overall Rating:** 4

**Summary:**

The authors mainly propose a pipeline for detecting, extracting, and anonymizing faces based on MTCNN and FaceNet. They also provide a crowdsourcing platform and invite users to correct false face detections (both false positives and false negatives) as well as to upload new face images and their annotations using the platform with an incentive of shared royalties generated from the commercial use of models trained using the data in the platform. With such a platform, the authors expect to amass a more "balanced" dataset (i.e., equally frequent faces across demographics) with fewer mislabeling with respect to gender or ethnicity that allows training models with less disparate performance across genders or racial groups.

**Strengths:**

- The authors address an important issue with high relevancy to the workshop theme
- The authors provide a relatively clear description of the proposed pipeline
- The results (Table 1 and Table 3) show an appealing improvement (less disparate accuracy, FNR, and PPV metrics across racial groups), highlighting the importance of using a more balanced dataset for model training.

**Weaknesses:**

- The focus is rather unclear and worsened by a less systematic introduction. While providing a "more" comprehensive solution is good overall, it would require the author to describe their proposal clearly and more systematically to avoid confusion.

- The significance and effectiveness of the idea are questionable. While the idea of providing a crowdsourcing platform is promising, its effectiveness is still questionable and can backfire. It is also not straightforward to see how the incentives scheme would result in a more balanced dataset, as it could be the case that some groups eventually provide more data vs. others. More importantly, the results (Table 3) appear to be not obtained using datasets from the proposed platform but instead using an already available dataset. If training using the FairFace dataset can already provide appealing improvement, what makes the author believe that providing another platform would yield an even more significant improvement? The premise of "retraining the face detectors in a periodic manner" also could not guarantee monotonic improvement and might cause various problems in redistributing the trained models. These issues, among many others, should be addressed to maximize the potential of the proposed idea.

- The authors use the term "bias" quite often without clearly defining its meaning in the context of the paper. Terms used with "bias" is therefore quite confusing, e.g., "inherent bias in these datasets," "mitigate the bias," "estimating inherent bias," or "estimating bias in the algorithms."

- The writing (especially word choice in many places) can be improved. E.g., "performance on images across the world" (pg. 2, line 40-41).

**Overall Recommendation:**

The current paper presents an incomplete solution that the authors try to tackle. While the presented problem of addressing "bias" is important and the proposed idea is interesting, the authors would need to showcase its effectiveness in tackling the problem (or highlight the challenges it faces if it does not work as effectively). The writing of the paper can also be significantly improved for the paper to be more readily presentable at the workshop.

**Review Confidence:**

4: The reviewer is confident but not absolutely certain that the evaluation is correct

---

### Decision · Program_Chairs · 2022-10-23

**Decision:**

Accept

**Comment:**

This submission is accepted based on its contribution in identifying the bias issues in face detectors from the data perspective. However, as the reviewer ts9S points out, the submission should provide more justification and evaluation in terms of how the crowdsourcing platform incentivizes the creation of a balanced dataset, especially with a quantitative study of the proposal in Section 3.3. Also, the use of the term "bias" should be more accurate. We strongly encourage the authors to solve these issues in the camera-ready version.